# ClimeApp: Data processing tool for monthly, global climate data from the ModE-RA paleo-reanalysis, 1422 to 2008 CE

Richard Warren[1,2], Niklaus Bartlome[1,2], Noémie Wellinger[1,3], Jörg Franke[2,3], Ralf Hand[2,3], Stefan Brönnimann[2,3], and Heli Huhtamaa[1,2]

[1]Institute of History, University of Bern, Bern, CH-3012, Switzerland
[2]Oeschger Centre for Climate Change Research, University of Bern, Bern, CH-3012, Switzerland
[3]Institute of Geography, University of Bern, Bern, CH-3012, Switzerland

*Correspondence to*:  Niklaus Bartlome (niklaus.bartlome@unibe.ch)
                                 or Richard Warren (richard.warren@unibe.ch)

**Abstract:** ClimeApp is a newly developed web-based processing tool for the state-of-the-art ModE-RA paleo-climate reanalysis. It presents temperature, precipitation and pressure reconstructions with global coverage and monthly resolution for the period 1422 to 2008 C.E. These can be visualized as maps or timeseries and compared with historical or other climate-related information through composite, correlation and regression functions. ClimeApp allows access to three data sets: 1. ModE-RA, a reanalysis that is created by assimilating early

instrumental documentary and proxy data into an ensemble of climate model simulations. 2. ModE-Sim, the native version of the underlying ensemble simulations, i.e. prior to data assimilation. 3. ModE-RAclim, an alternative version of the reanalysis product. Together, these allow researchers to separate the effects of model simulations and observations on the reanalysis. The app is designed to allow quick data processing for climatologists and easy use for non-climatologists. Specifically, it aims to help bring climate into the humanities,

where climatological data still has huge potential to advance research. This paper outlines the development, processing and applications of ClimeApp, and presents an updated analysis of the calamitous Tambora volcanic eruption and the 1816 'year without a summer' in Europe, using the new ModE datasets.
ClimeApp is available at https://mode-ra.unibe.ch/climeapp/.

## 1 Introduction

Interdisciplinary research is a great facilitator of scientific progress. It allows researchers to address all aspects of a problem and take a holistic view not limited to one specialised field. To assess, for instance, the impact of a volcanic eruption such as Mount Tambora in 1815, volcanologists may study the eruption itself (Kandlbauer and Sparks, 2014), climatologists use numerical models to assess the climate impact (Raible et al., 2016), historians look into administrative records for social impacts (Krämer, 2015), and economists and epidemiologists look for

effects on trade and the spread of disease (Wood, 2014). By combining their results, researchers can obtain unique insights relevant to all fields, as demonstrated by Brönnimann and Krämer (2016). In recent years, Tambora research has focused on the climate-society nexus, investigating the vulnerability and resilience of societies affected by the climatic shock of the eruption. The global climate anomaly caused by Tambora significantly impacted crop yields and harvests, the subsequent impact on human societies was modulated by

socio-economic and political factors. Since such complexity requires expertise from many different fields, interdisciplinary research is therefore both relevant and necessary for understanding how societies cope with such extreme events (Flückiger et al., 2023).

To bridge disciplines as different as history and climate science, it is crucial to make data accessible to non-specialists. ClimeApp is a web-based application that achieves this by giving researchers without a climatology

background the possibility to add climate reconstruction data to their sources and analysis. It facilitates quick and easy processing of the ModE-RA climate reanalysis and associated datasets (Valler et al., 2024), allowing both climatologists and other researchers to evaluate the data without time-consuming coding. Functions for both casual investigation, as well as detailed statistical and source analysis, are built in. It is also possible to upload your own research – be it prices, harvest yields or mortality – as a timeseries to compare with the ModE-RA

dataset. ClimeApp, developed using the Shiny R package (Chang et al., 2024), stands out for its simplicity and accessibility in presenting and processing complex data. This paper summarises the ModE-RA, ModE-Sim and ModE-RAclim datasets used in ClimeApp and outlines the main features of ClimeApp's internal structure and external interface. We then examine the 1815 Tambora eruption and the following 'year without a summer' in Europe to showcase the functions and applications of both ClimeApp and the ModE data. This demonstrates how

reanalysis data can be used to distinguish between internally and externally forced variability of the climate system. We conclude by expanding on the app's potential for both the humanities and climate sciences. Detailed reference material for ClimeApp data processing can be found in the appendices.

(a)

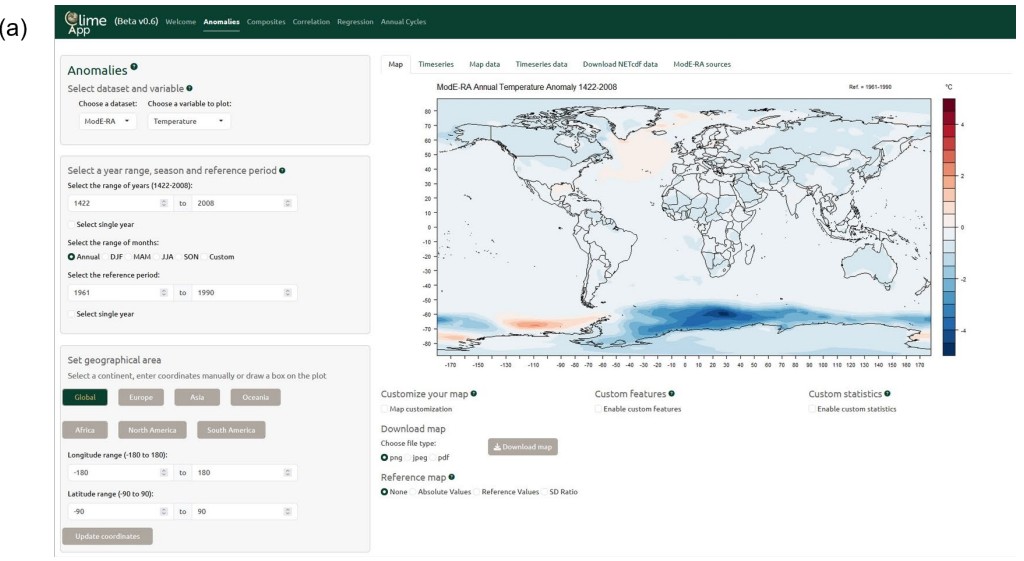

(b)

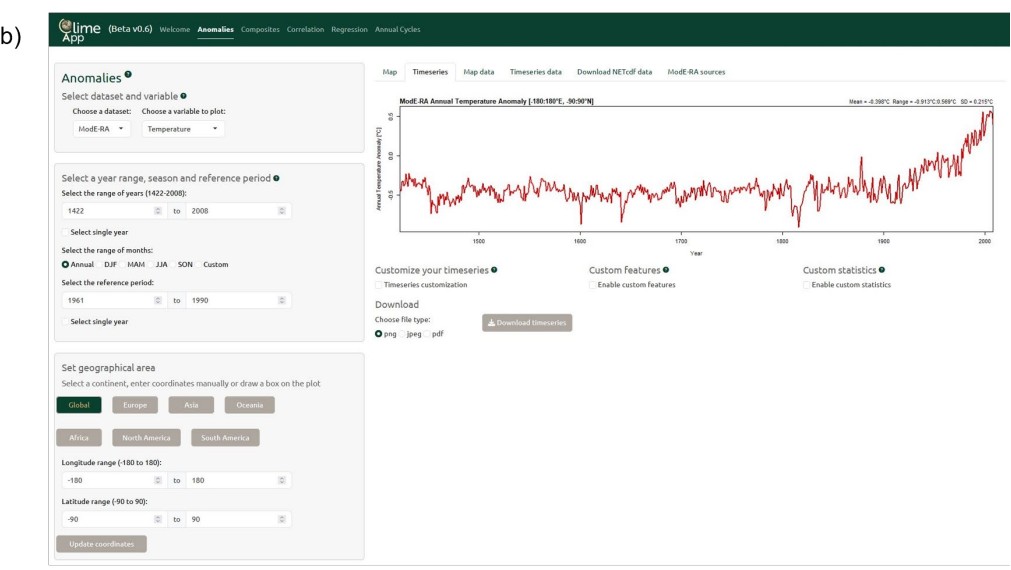

(c)

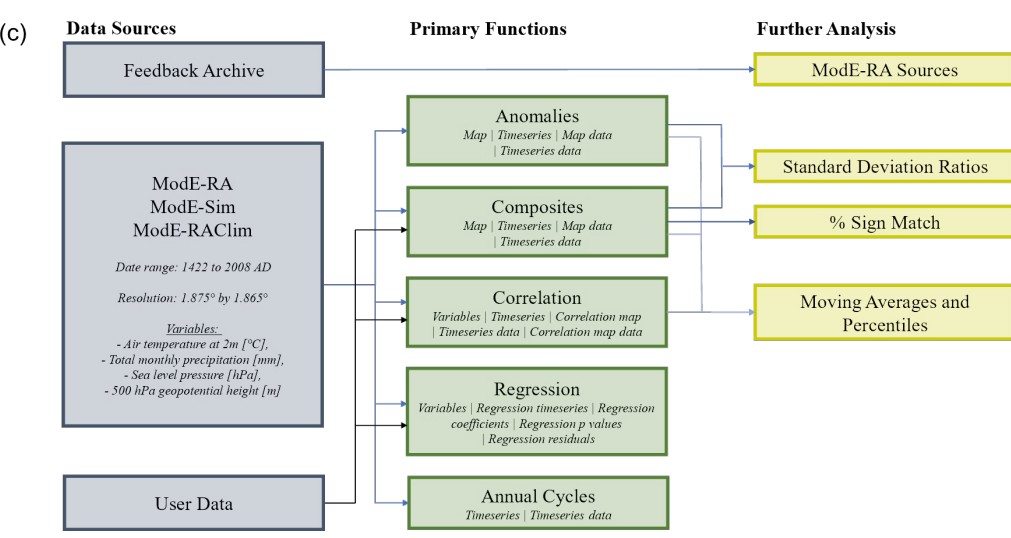

**Figure 1.** Layout, functions and data flow of ClimeApp. (a) ClimeApp, displaying a global temperature anomaly map. (b) ClimeApp showing the same global temperatures as a timeseries. (c) Summary of the data sources used by ClimeApp, along with its five primary functions and current options for further data analysis.

## 2 Data Sets

### 2.1 ModE-Sim

ModE-Sim is an ensemble of global climate simulations, driven by external forcings. The ensemble is produced by inputting reconstructed volcanic aerosols, total solar irradiation and greenhouse gases into the ECHAM6 atmosphere general circulation model (Stevens et al., 2013), with observed/reconstructed sea surface temperature (SST) and sea ice as boundary conditions. The different ensemble members are produced by using different versions of the SST and volcanic forcings, which accounts for the uncertainty in the reconstructions. Each

member in the ModE-Sim ensemble therefore represents a possible climate state that is physically compatible (from the model's perspective) with the prescribed external forcings and boundary conditions (Hand et al., 2023). The ModE-Sim ensemble, which is the basis/prior for ModE-RA (see below), is therefore a combination of externally forced signals and internal variability predicted by the model physics. Note, however, that ModE-Sim is an atmosphere-only simulation, so feedbacks between the ocean and atmosphere are not included. The

ModE-Sim ensemble mean, used by ClimeApp, is the average over all ensemble members. Averaging tends to "smooth out" most of the internal variability, but does retain signals caused by external forcing, e.g. the climate's reaction to a volcanic eruption. On its own, ModE-Sim can be used as a tool to study climate variability, but in combination with ModE-RA it can also help climatologists identify how observations affect the final reanalysis (see our case study, section 4.1).

### 2.2 ModE-RA


At the time of writing, the Modern Era Reanalysis (ModE-RA) is the most comprehensive reconstruction of the monthly global climate of the past 600 years (Valler et al., 2024). By assimilating a huge database of climate observations, each ModE-Sim ensemble member is corrected/updated towards the observations, weighted by individual uncertainties in both observations and simulations. The number of observations increases rapidly

through time, rising from a few thousand natural proxies and historical documents in the 15[th] century to over 100,000 (mostly instrumental) measurements by the early 20[th] century. These climate observations are assimilated biannually for each year. Natural proxies include tree rings, ice cores, corals, speleothems and lake sediments, while historical proxies are gathered from weather diaries, phenological data and early ship records. Where and when records are available, this allows observation-based reconstruction of all four seasons,

improving on previous studies which mostly focussed on the summer season. If no observations are available, ModE-RA is identical to ModE-Sim. Note that the ensemble mean of ModE-Sim mainly represents the model's response to external forcings, but the assimilation of observations into ModE-RA brings back internal variability. This also means that there is increasing variability in the ModE-RA ensemble mean over time (as the number of observations increases). The spatial horizontal resolution for ModE-RA is 1.875° (longitude) by 1.865°

(latitude), while the temporal resolution is one month.

### 2.3 ModE-RAClim

ModE-RAclim (Valler et al., 2024) is an alternative version of ModE-RA, designed to focus on the observations, by minimising its input from the ModE-Sim climate models. In ModE-RA, observations are assimilated into time-aligned ensemble members from ModE-Sim (i.e. for the year 1800, the basis for ModE-RA is the externally

forced ModE-Sim states for 1800). By contrast, ModE-RAclim uses the same offline assimilation process (see

Valler et al., 2024 for more detail), but using non-time-aligned ModE-Sim members. Instead, for each year, a random sample of 100 years from any of the ModE-Sim ensemble members is used to form the ModE-RAclim ensemble, prior to assimilation. This means that in ModE-RAclim, the externally forced signal in the model simulations is removed from the ensemble and only added back if it appears in the observations. Hence,

following a volcanic eruption, ModE-RA and ModE-Sim will both include the model response to stratospheric aerosol forcing, but any volcanic signal in ModE-RAclim will exclusively result from the observations. ModE-RAclim is therefore useful for distinguishing whether observed climate anomalies are a consequence of the external forcing that was used for the model simulations or a result of data assimilation (see our case study, section 4.1).

**3 Interface**

**3.1 Building a Shiny app – User Interface (UI), Server and Helpers**

R Shiny Apps have become popular not only for research in general (Gebauer et al., 2023) but particularly in climate science. They provide a powerful set of tools for quickly creating and deploying simple applications for presenting or processing data. Other scientific applications designed using R Shiny are described in: (Sousa,

2019; Fajardo et al., 2020; Möller et al., 2020). There are three main components to a Shiny app: The local server where all the necessary data is stored, the R environment and the web interface.

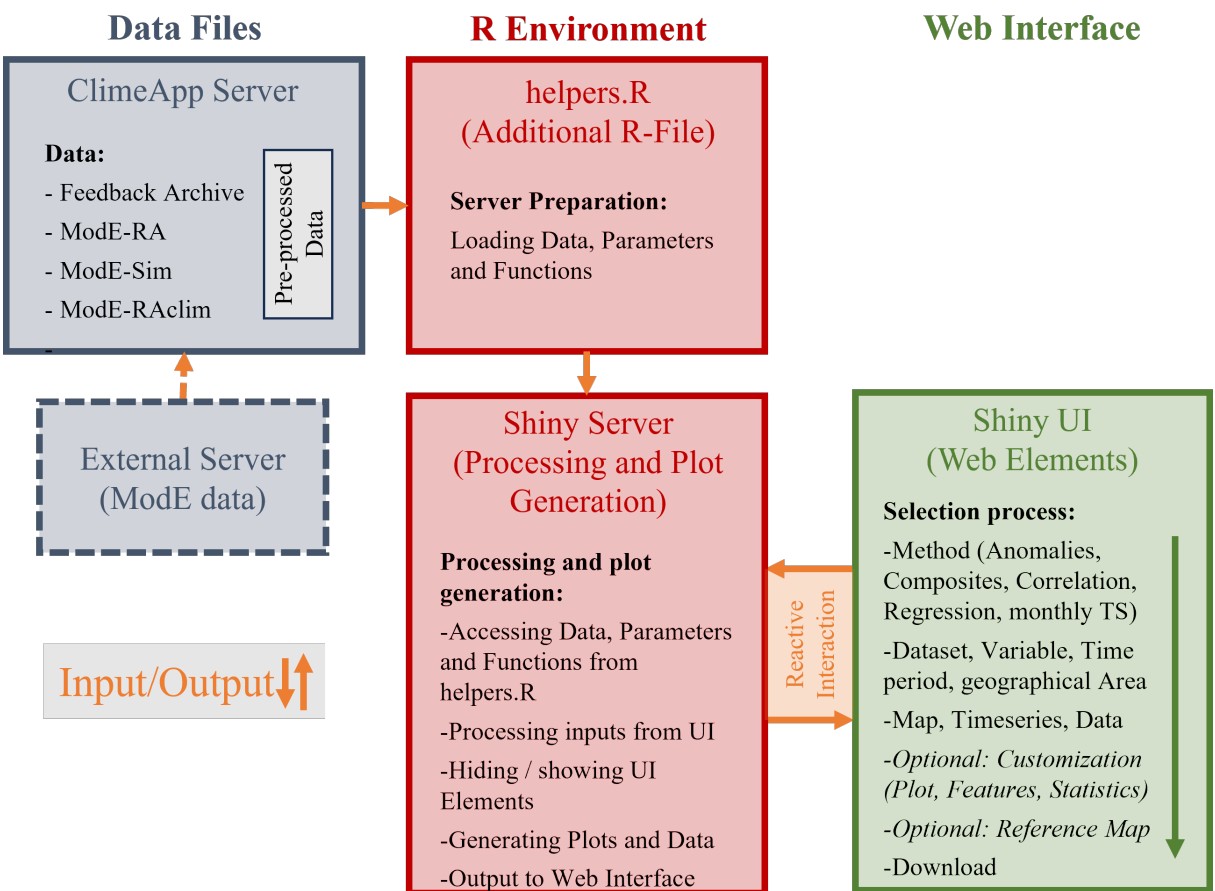

**Figure 2.** Internal Structure of ClimeApp, showing how the data storage, R environment and web interface interact.

In ClimeApp, ModE data from an external server is stored locally on the ClimeApp server, including some pre-processed data to reduce loading times. All the server preparation, such as loading the R-libraries, unpacking the data, loading in parameters and predefined functions is done in a supporting R-file called *helpers.R*. The main *app.R* file consists of a *UI* section, where all the elements are defined, named and positioned, and a *Server* section where processing of inputs and plot generation is performed. When in use, the *Server* and web interface

interact with each other reactively, with inputs from the web elements processed in real time and then output as plots or data tables. All web elements have preset inputs to generate an initial plot, which can then be altered by the user.

## 3.2 Usability

The ClimeApp web interface is designed to streamline data selection, providing all necessary information and

options without being overly complex. The initial *Welcome* page displays general information about the ModE database and ClimeApp and gives access to five main tabs, one for each of the primary functions shown in Figure 1. We deliberately decided to supply only functions that are commonly used in historical climatology, but allow users to easily download selected ModE data for further processing. Input and customization options are selected within a tab, appearing on the same screen as the plots. For new users, it is vital not to overload the

interface with information - be it with text or advanced options. To avoid this, we work with help texts that explain how plots and data are calculated, but only when selected. The interface is also streamlined by hiding many UI elements until they are required. Non-standard tools such as custom months, single year maps and additional features to customize plots are only visible when the relevant customization option or tick box is selected.

## 3.3 Customization

Having generated a plot, users have several options to customize their maps and timeseries. ClimeApp is designed to create graphics suitable for publication without the need for further editing in other programmes. The app supplies three sets of tools to customize plots: *Customize your map/timeseries*; *Custom features;* and *Custom statistics*. Under *Customize your plot*, the user can change titles, subtitles and the plot's axis. Once selected, axis

values will stay fixed even after a plot has been changed, making it easy to quickly compare different regions and time periods. *Custom features* adds and removes points, lines and highlights to/from the plot and includes a location search function (Tennekes, 2021) that can add labelled, geocoded points. *Custom statistics* allows users to add the *SD ratio* or *% sign match* statistics (see Appendix B for more information) as overlays to their maps or add percentiles and moving averages to their timeseries. All customization stays, even when plots are altered or

different variables selected, saving time when creating multiple plots. Customization and plot inputs can also be downloaded, saved and re-uploaded to quickly recreate previous plots.

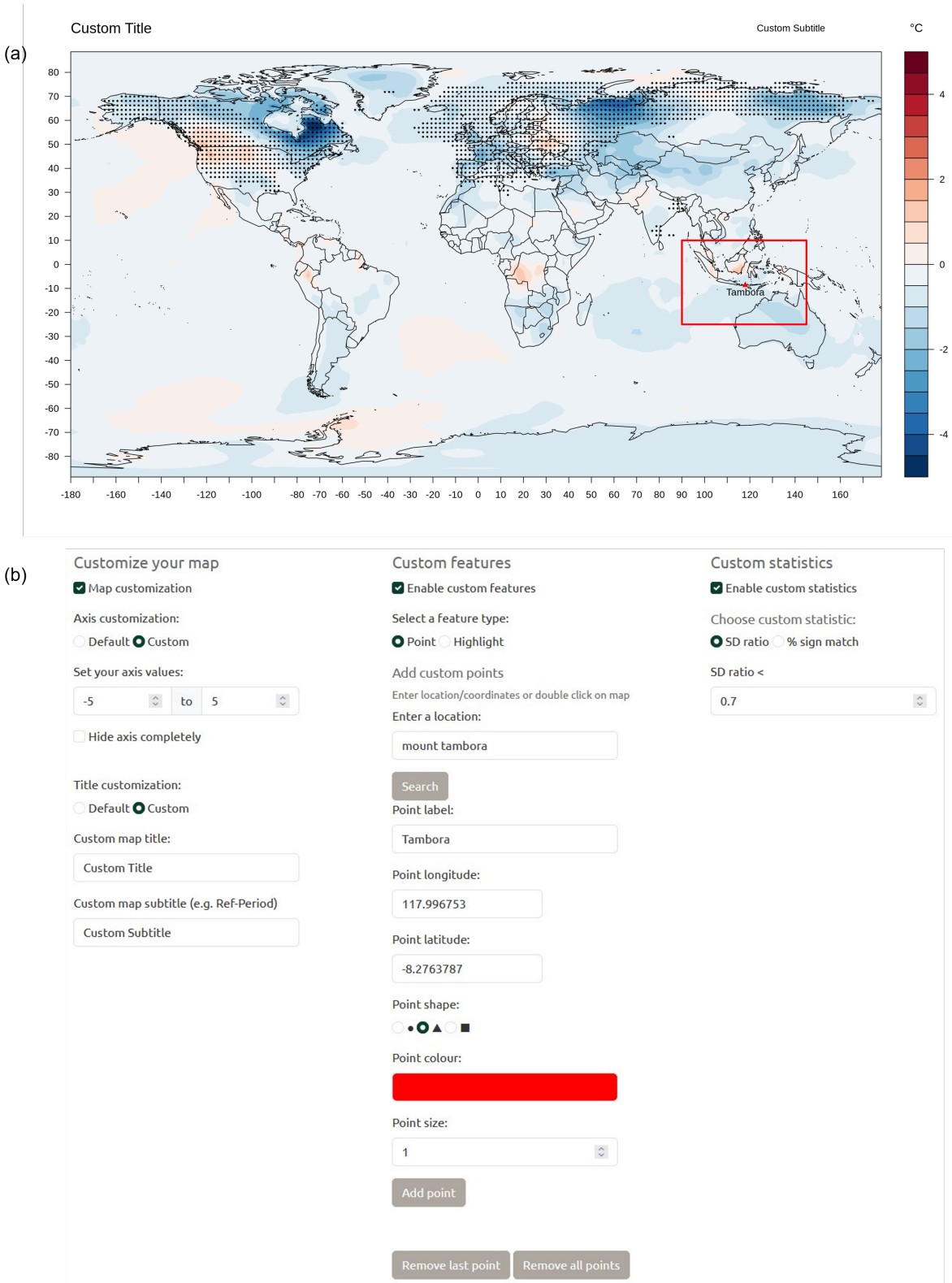

**Figure 3.** Example of a customized map from ClimeApp, showing: (a) An anomaly map using several custom features (customized title, custom axis, labelled point, highlight box and statistical overlay showing regions that match a chosen criteria); (b). ClimeApp's map customization options.

### 3.4 Uploading/downloading data and plots

Nearly all the plots in ClimeApp can be downloaded in multiple file formats. Plot data can also be exported for use in other programmes. In the *Anomalies* tab, advanced users can download multiple ModE variables as a

single NetCDF file, while in the *Regression* tab, a statistical summary of the linear regression is also available for download. Users can upload their own data in three of the main tabs: *Composites* can be created from an imported set of years, while tables of annual data can be uploaded for *Correlation* or *Regression* analysis. Uploaded files can be compared with either ModE variables or further user data, making ClimeApp a useful tool for performing simple correlation and regression analysis on your own material. Since files must be correctly formatted for ClimeApp to understand them, when uploading, users are shown an example image demonstrating how to correctly structure their file.

**3.5 Reactive functions and pre-processing**

One major benefit of Shiny apps is instant updating – as soon as a user selects new values, ClimeApp will automatically update its plots to match the new selection. This is timesaving and allows researchers to explore the climate data quickly and efficiently. Exceptions have been made for changing the geographical area, adding points, lines and highlights and using *Annual Cycles* function, to allow all information to be entered before updating the plot. To enable plots to instantly update, ClimeApp uses R Shiny's *reactive functions*. These functions track all the variables they depend on, re-executing if one of these variables changes. This means that only sections of code where a change has been made are re-run, reducing processing power and increasing response time. Reactive functions chain - each passing an output to a further reactive function until the final reactive output is passed to the UI, updating the onscreen plot or table.

To further reduce processing time, ClimeApp employs pre-processed data for common user selections. These include five seasonal averages: December-January-February (DJF); March-April-May (MAM); June-July-August (JJA); September-October-November (SON); and Annual; and four variables: Temperature; precipitation; sea level pressure (SLP); and 500 hPa geopotential height (Z500). Annual means for each are already calculated and stored on the ClimeApp server, allowing them to be plotted without further computation. This considerably reduces the memory and time required for data processing and is especially valuable when users are working on large geographical areas or over long time periods.

**4 Case Study – Tambora reanalysed**

In 1815, on the Indonesian island of Sumbawa, Mount Tambora erupted. Large volumes of sulphur were injected into the stratosphere, forming a layer of aerosols around the globe. This layer considerably reduced incoming solar radiation, significantly affecting the global and particularly European climate (Brönnimann and Krämer, 2016). The study of the Tambora eruption's aftermath is a prime example of interdisciplinary research. Already in the 19[th] century, research fields such as meteorology, geology, and natural history were cooperating to study the eruption. Despite limitations in data, expertise, and analytical tools linking climate anomalies to volcanic activity, this research helped to advance existing theories, such as the Ice Age hypothesis (see Brönnimann and Krämer, 2016). In more recent times, scientists from the historical disciplines, climatology, aerosol science, geology, epidemiology and many other subjects have all combined their efforts to more fully comprehend the mechanisms and global impacts of such major volcanic eruptions (Wood, 2014). Today, using the latest and most comprehensive climate data (ModE-RA, ModE-Sim and ModE-RAclim) and the tools in ClimeApp, we can continue this research, taking a new look at the eruption's impact.

**4.1 Tambora in ModE-RA, ModE-Sim and ModE-RAclim**

The year of 1816, following Tambora, has become known as the 'year without a summer', due to the extreme global temperature anomalies that followed the eruption. Using ClimeApp's *Anomalies* function, we can see the unusually cold summer temperatures over Europe in all three ModE datasets:

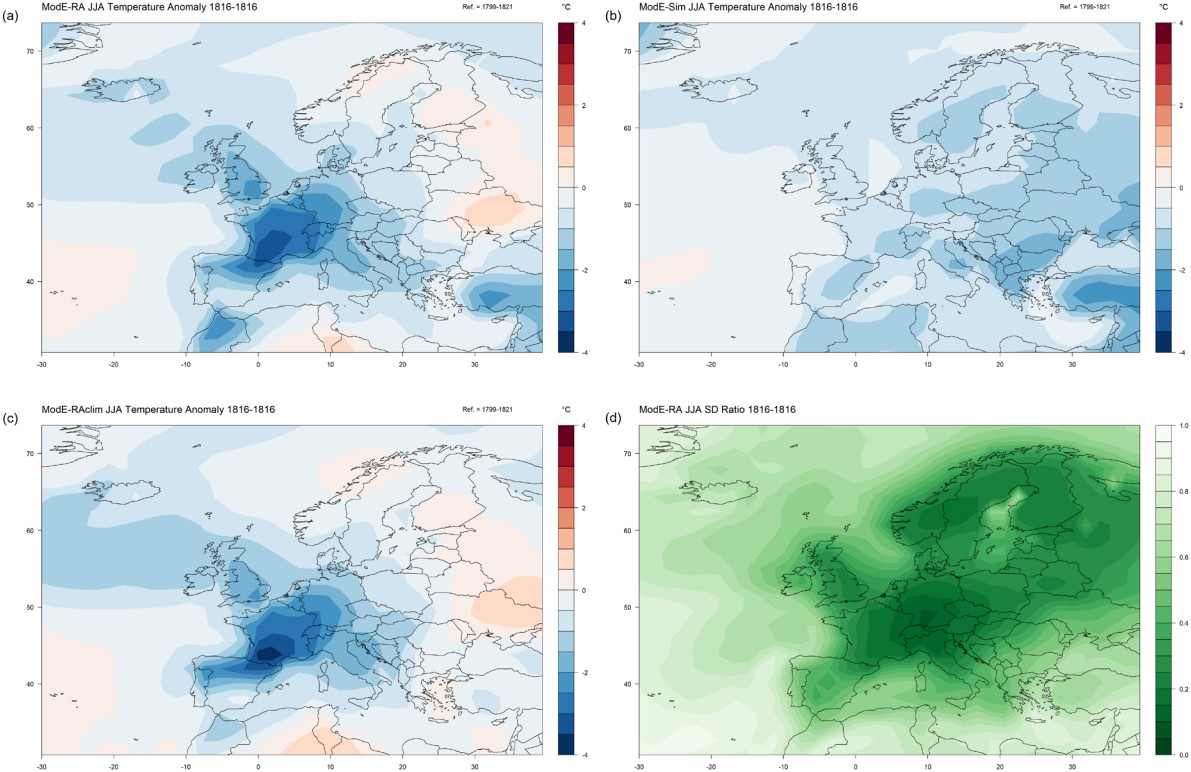

**Figure 4.** (a), (b) & (c) European temperature anomalies for boreal summer (June to August, JJA) 1816, as compared to reference period 1799 to 1821. Showing results from ModE-RA, ModE-Sim and ModE-RAclim, respectively. (d) The standard deviation ratio (or SD ratio) between the ensembles of ModE-RA and ModE-Sim (see Source Analysis and Further Statistical Functions in Appendix B for more information).

A comparison between the datasets shows that the 1816 anomalies in ModE-RA and ModE-RAclim are essentially the same, while ModE-Sim reports considerably smaller temperature changes. This suggests that either certain mechanics of the Tambora eruption are not captured by the ModE-Sim ensemble, or that only part of the observed cooling can be explained by aerosol forcing. Evidence for the latter can be found in SLP and Z500 data from ModE-RA (see Appendix C). These show a significant low-pressure anomaly over western Europe in 1816, which would have further cooled air temperatures. It may have also simultaneously caused the positive temperature anomalies over northeastern Europe, as an anticyclone around the low-pressure system drew warm air northwards from the eastern Mediterranean. The absence of low pressures in ModE-Sim (see also Appendix C) suggests that these 'additional' temperature changes are largely due to internal climate variability not captured by the ModE-SIM ensemble mean. This would support the suggestion by Auchmann et al. (2013) and Brönnimann and Krämer (2016) that additional cooling in the 1810s may have been attributable to the long-term effects of the 1809 'unknown eruption', low solar activity and internal decadal variability in the ocean-atmosphere system (though it would suggest that solar activity might have played a less significant part).

The similarity between ModE-RA and ModE-RAclim results from the profusion of data sources for this period and region - 4778 European observations were assimilated for April to September in 1816 alone (see Figure 6).

These constrain ModE-RA and ModE-RAclim, but not ModE-Sim. We can see this from the SD ratios in Figure 4d. The SD ratio measures the difference in spread between the ensembles of ModE-RA and ModE-Sim. An SD ratio of 1 means that both ensembles have the same standard deviation, i.e. nothing was assimilated or observations had no effect, while an SD ratio of 0 indicates an overconfident reanalysis since no uncertainty would remain. In general however, a lower SD ratio indicates where more information has been assimilated. In 4d, we can see this over central and northern Europe, where the lower SD ratio suggests a more reliable reanalysis, or at least one that is closer to the observations.

However, the SD ratio should not be the only measure of the dataset's reliability. If we use ClimeApp to view global precipitation rather than temperature in 1816, we can see the disparity between the ModE datasets reverse:

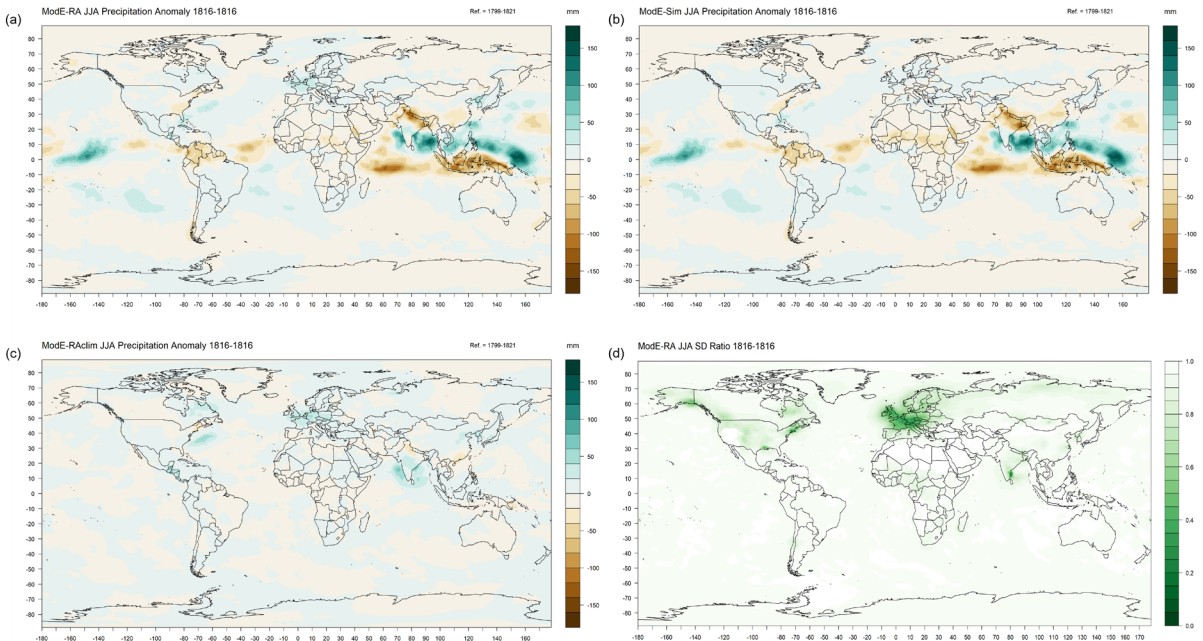

**Figure 5.** (a), (b) & (c) Global precipitation anomalies for June, July and August 1816, as compared to reference period 1799 to 1821. Showing results from ModE-RA, ModE-Sim and ModE-RAclim respectively. (d) Standard deviation (SD) ratio of ModE-RA ensemble to ModE-Sim ensemble, showing significant constraint by observations only over Europe, coastal North America and SE India.

Here, ModE-RA very closely matches ModE-Sim, but not ModE-RAclim. This is because, while the tropical monsoon should have been affected by Tambora (Marti and Ernst, 2009), there are few equatorial observations to capture this. ModE-RAclim is therefore unable to reconstruct significant changes in tropical precipitation. ModE-Sim meanwhile, includes physical mechanisms predicting extreme monsoon changes from volcanic aerosol forcing, and passes these on to ModE-RA. Interestingly, in the one tropical location that is more constrained - southern India (see Fig. 5a) - ModE-RAclim does more closely match the predictions from ModE-Sim. Results from both precipitation and temperature broadly agree with previous reconstructions (Brönnimann, 2015, Fig. 4.23) and modelling (Wegmann et al., 2014, Figs. 3 and 4a). There are some regional differences though, such as the absence of eastern European warming in Wegmann et al. (2014), and the presence of eastern European drying in Brönniman (2015) (as well as considerable cooling over Norway and Sweden) which probably reflect improved modelling and more comprehensive observations.

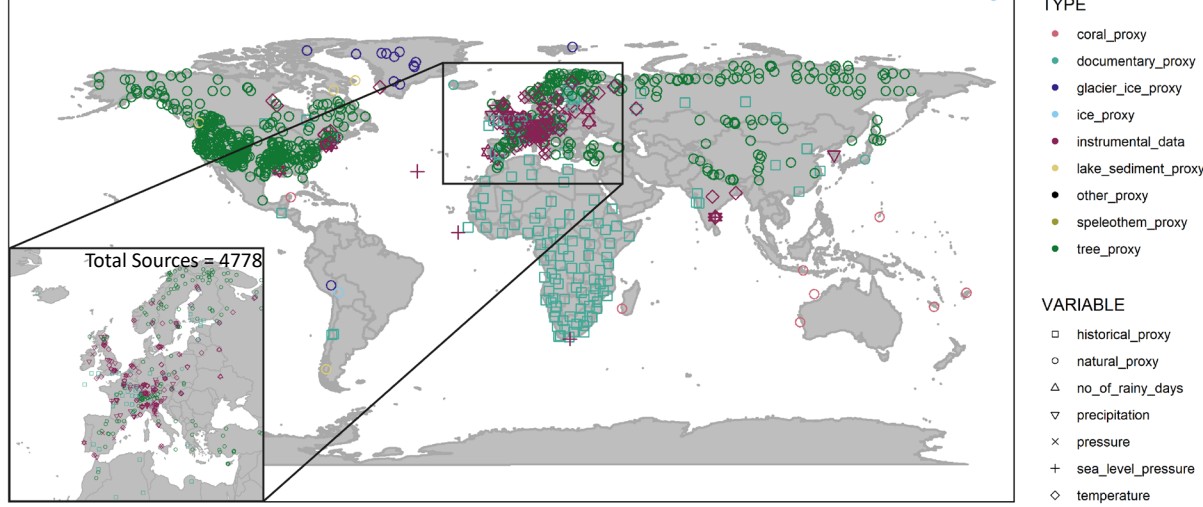

**Figure 6.** ClimeApp *ModE-RA sources* plot, showing all assimilated observations for April to September 1816. By comparing the sources plot to the SD ratios (Figs. 4d and 5d), we can see which observations significantly constrain the ModE-RA ensemble for a particular variable. For example, instrumental data (red) noticeably constrains JJA temperatures in Europe (Fig. 4d), but the documentary proxies (yellow) in Africa have only a minor effect on the ensemble (possibly because the proxies correlate only weakly with boreal summer temperatures).

## 4.2 Compositing volcanic years

To compare Tambora with other volcanic eruptions, we can use the *Composites* function in ClimeApp to view the temperature anomalies in 1816 alongside a composite of the anomalies following other major eruptions:

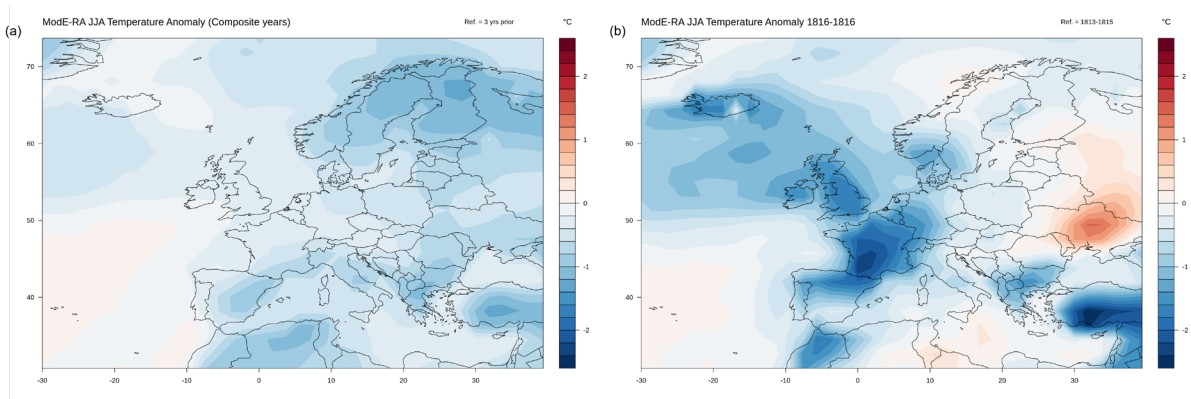

**Figure 7.** (a) Composite of JJA temperature anomalies following the 12 largest volcanic eruptions, excluding Tambora, between 1422 and 2008, as measured by their impact on global stratospheric aerosol optical depth (Toohey and Sigl, 2017). Composited years are 1454, 1459, 1596, 1601, 1642, 1696, 1784, 1810, 1832, 1836, 1884 and 1992, corresponding to the maximum volcanic forcing following each eruption. Anomalies are calculated with respect to the three years preceding each composite year. (b) Temperature anomalies in 1816, following Tambora, as compared to the three years prior.

In the ModE-RA dataset, we see considerably larger anomalies following Tambora than in the more general composite of other eruptions. While this could suggest other, non-volcanic effects on the European climate, it is likely that the more individual patterns of each eruption are smoothed out in the composite, generating the more homogeneous weaker cooling. In this case, compositing highlights the forced part of the volcanic signal, similar to running an ensemble of simulations. This would explain the similarity between Fig. 7a and Fig. 4b (the ensemble mean from ModE-Sim). The disparity may also partly result from the volcanic forcing following Tambora being approximately twice that of the composite sample or the fact that the composite volcanic years were also generally less constrained than 1816 and therefore closer to ModE-Sim.

**4.3 Removing the volcanic signal?**

Using ClimeApp, we can also attempt to remove the volcanic signal altogether from ModE-RA. The volcanic forcing data used in ModE-Sim (Toohey and Sigl, 2017), measures global volcanic forcing through reconstructed stratospheric aerosol optical depth (SAOD). ClimeApp's linear *Regression* function can build a simple statistical model linking SAOD to European JJA temperatures. In principle this can 'remove' associated effects (at least according to the linear regression model), leaving the 'residual' temperature variation unrelated to volcanic forcing (Fig. 8c):

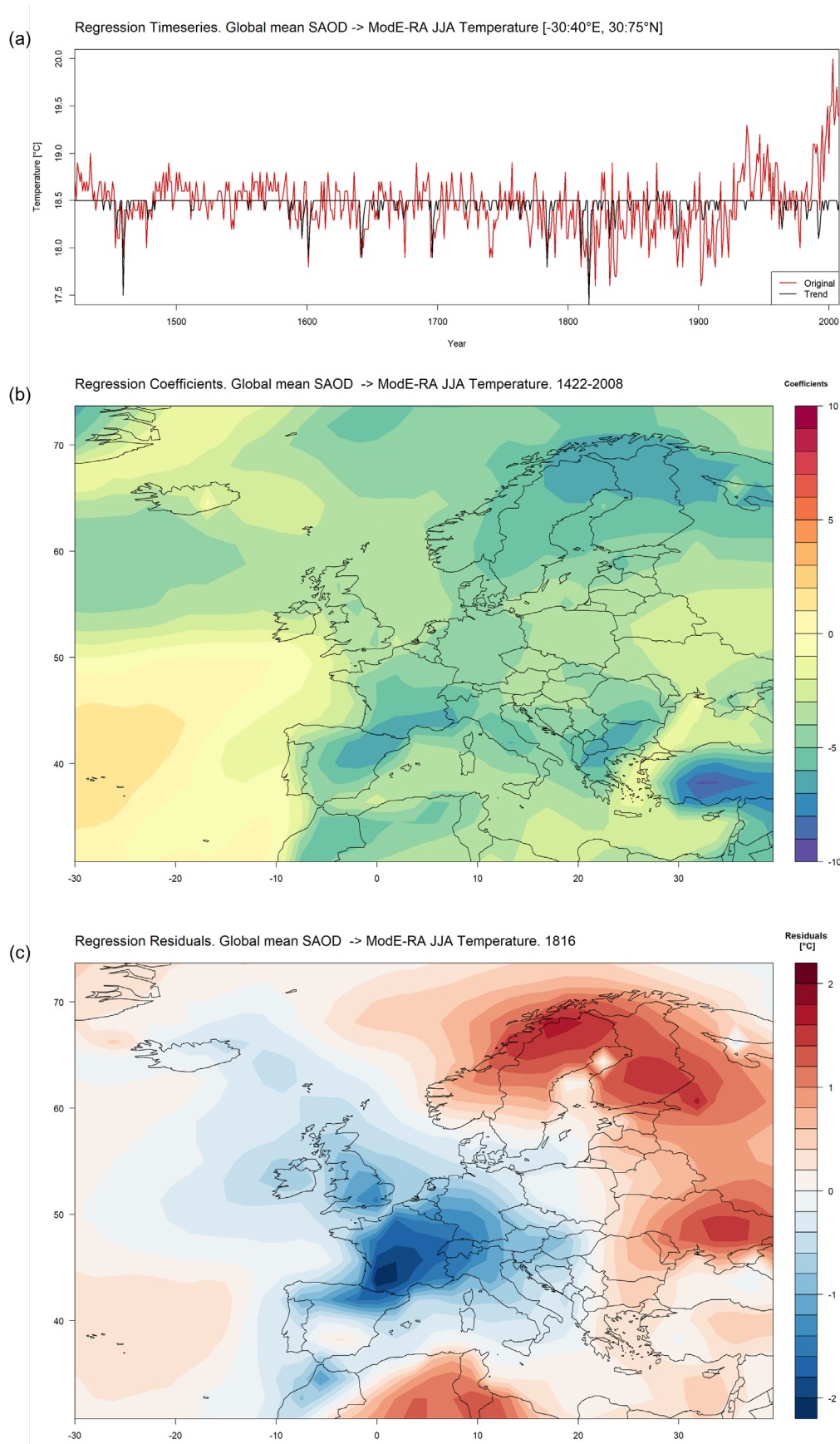

**Figure 8.** (a) Timeseries showing the 'original' European JJA temperatures (1422-2008) used to train the model and the predicted 'trend' in temperatures from the regression model. Note that both timeseries show the spatial average for each year, while the actual regression model consists of an original and trend for each point on the map. (b) Map showing the linear regression coefficients linking JJA temperature and global SAOD. (c) Residual European JJA temperatures in 1816.

The residuals (Fig. 8c) show significant non-volcanic anomalies in 1816, that generally match the results from the anomaly and composite analysis. We can see the additional cooling over western Europe and concurrent warming of eastern Europe that *may* be associated with the low-pressure system found in ModE-RA. The regression coefficients (Fig. 8b) agree with the results from ModE-Sim, predicting a mostly homogenous cooling over the European landmass in response to aerosol forcing. However, any further conclusions are limited by the

assumption of a linear relation between SAOD and JJA temperatures. This is unlikely, given the complex atmospheric dynamics governing the climate response to the radiative forcing (Brönnimann, 2015). Furthermore, the limited number of major eruptions reduces the reliability of any relationship drawn from a simple comparison of the data.

**4.4 Comparing ModE-RA to historical data**

Tambora didn't only affect the climate, human society was also significantly impacted. At the turn of the 19th century, grain prices were the most important indicator to measure the state of an economy (Ljungqvist et al., 2022). Even if other factors such as demand, military conflicts and the quality of the harvest played a role, quantitative supply was an important determinant of price (Krämer, 2015). We would therefore expect a link between price data and temperatures during the growing season. This can be tested using a sample of bread price

data from Lucerne, Switzerland and ClimeApp's *Correlation* function:

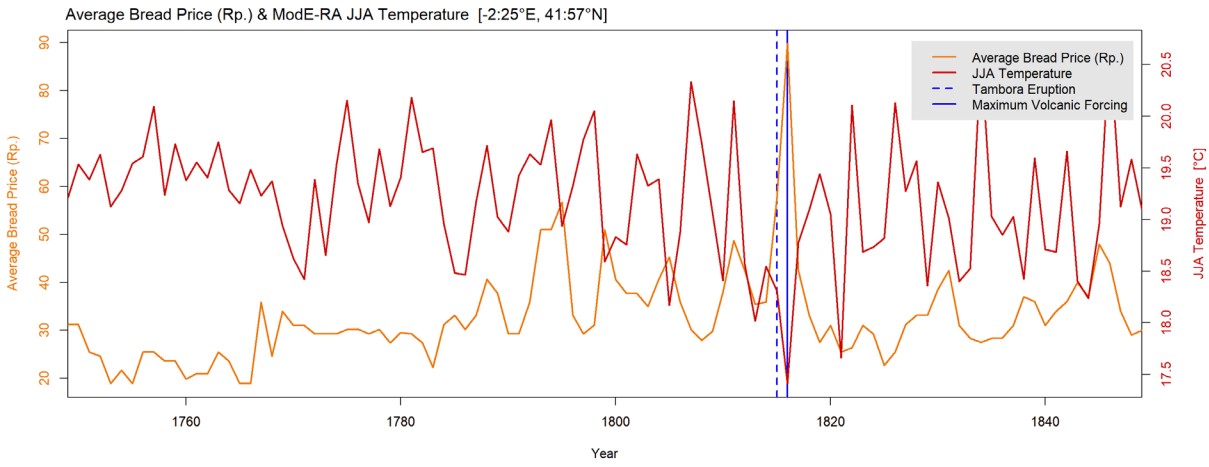

**Figure 9.** Timeseries correlation showing JJA temperatures from 1750-1850 over central Europe and the average price of bread ("Weissbrot") in Lucerne, Switzerland for the following year (Haas-Zumbühl, 1903). Timeseries correlation of r = -0.328, with a p value < 0.01. Prices are in Rappen per kg. Note that the following year bread price was used due to the lagged effect of

temperature during the growing season on market prices.

The timeseries correlation gives us an inverse correlation of r = -0.328. This implies a weak but general correlation between the growing season temperatures and grain prices, where low temperatures presumably led to a poor season and a spike in bread prices the following year. The connection is particularly visible after Tambora, where we see both a sharp rise in prices (in 1817) and a sudden temperature drop during the summer

season of 1816. This coincided with unusually high precipitation, which may explain why we do not see a similar price spike in 1813 or 1821 when precipitation was moderate. Our results agree with previous work on Tambora, which demonstrated how adverse weather in Switzerland in 1816 played a crucial role in massively reduced crop yields (Flückiger et al., 2017). In the following year, this led to an increase in prices for grain and bread and widespread famine, particularly in eastern Switzerland (Brönnimann and Krämer, 2016).

## 5 Outlook

### 5.1 Potential for humanities

It is well known how intertwined climate and society have been over the last 1000 years (Pfister and Wanner, 2021). Naturally, this has important implications for all sciences that concern themselves with humanity, but especially for history. An understanding of the climate can complement traditional historical approaches, adding a crucial dimension to our understanding of the past. As demonstrated in our case study, anomaly plots for specific years can illustrate how different regions were affected by extreme climate events. Correlations can measure statistical connections between climate and historical variables, ranging from crop yields and mortality rates to price data. Knowledge of climate history can also give us perspective on the current climate crisis, with historical examples employed as case studies for the impacts of a changing climate on human societies (Lamb, 2002; Degroot et al., 2021). However, there are still challenges to interweaving the two disciplines. These include the heterogeneity of approaches, the diversity of disciplinary perspectives and often mutually unintelligible terminology (White et al., 2023). We can innovate to overcome these difficulties however. There exists a growing collection of databases and tools for accessing documentary sources and climate data for use in historical climatology. Some primary examples include tambora.org (Riemann et al., 2015), Euro-Climhist (Pfister et al., 2017) and Climate Explorer (Trouet and van Oldenborgh, 2013). Each tool fulfils a different niche. Tambora.org and Euro-Climhist both provide a vast number of original sources and indexed data from (predominantly) human archives. This qualitative data - usually documentary data, but in the case of Euro-Climhist also visual sources such as paintings – is extremely helpful for investigating and interpreting singular events or climatological trends. However, it is often sporadic and temporally and geographically inhomogeneous, hence the advantage of modelling and reanalysis products like ModE-RA and ModE-Sim. Climate Explorer meanwhile, is an excellent tool for accessing and analysing some older climate reconstructions, as well as modern instrumental data. However, its vast array of options and data can often be confusing for new users, and it does not give access to the ModE-RA data.

This is where ClimeApp is able to provide a complimentary tool, utilizing a smaller range of data and functions, but via a far more user-friendly interface. ClimeApp provides highly customizable plots, along with helpful explanations where necessary. It also, uniquely, allows users to view all the sources behind the reanalysis data. Combined with the standard deviation ratios (see case study, section 4.1), these allow detailed assessment of the reliability and applicability of the reanalysis. For historians, this possibility to make source-attributed, publication suitable plots without coding skills is invaluable. Furthermore, for researchers using quantitative data, ClimeApp can be a convenient tool for correlation or regression analysis without needing external programmes. However, the most important feature of ClimeApp is that it uses the most extensive and modern global reanalysis for the climate of the last 600 years, which can be directly compared with historical socio-economic time series. In general, the broad appeal of ClimeApp for the humanities lies in its simplification of accessing, visualising and analysing the latest reanalysis data.

### 5.2 Potential for climate sciences

We have already demonstrated some of ClimeApp's potential for climate science in our case study, but a few further points are worth making here. First and foremost, ClimeApp can save researchers considerable time

creating field and timeseries plots that would otherwise have to be manually coded. As it incorporates many
analysis methods used in historical climatology and paleoclimatology, climatologists studying the age before
instrumental measurements will find the ModE data invaluable. Our case study showed how contrasting ModE-
RA, ModE-Sim and ModE-RAclim can give constructive insights into the causes of certain climate anomalies,
while the source plots and SD ratio data can be used to assess the reliability and the limitations of ModE data.
For the developers of ModE-RA, ClimeApp is particularly useful for testing their data, allowing them to quickly
visualise and compare the datasets. Other researchers can use the app to assess new paleoclimate data, using the
*Correlation* function to map correlations between their data and ModE-RA. For students and lecturers, we hope
that ClimeApp makes advanced reanalysis data easily accessible for exercises and project work. Finally,
ClimeApp can be a useful template for developing other Shiny projects, particularly those for processing and
plotting complex data. The source code for the application is openly available (see Code availability) and can be
freely adapted for other applications.

**6 Conclusions**

This paper summarized the functionality and potential of the new ClimeApp web application. It demonstrates
how simple programs can be powerful tools to make specialized data available to all. The application provides
historians, climatologists and other researchers quick access to state-of-the-art ModE climate reanalysis.
Through a user-friendly interface, students and scientists can view and analyse the historical global climate with
just a few clicks. Furthermore, and uniquely, ClimeApp allows detailed investigation of the sources used in a
climate reconstruction. The *ModE-RA sources* tool displays the type and location of all proxies and documentary
sources used to constrain the ModE-RA climate models, while the *SD ratio* statistical analysis quantifies their
effect on the final data. Re-examining the Tambora eruption and the 1816 'year without a summer', we
demonstrated how ClimeApp can contribute to new scientific research. The tool was used to combine results
from a reanalysis (ModE-RA), an ensemble of climate models (ModE-Sim) and an observation-focused
reconstruction (ModE-RAclim), allowing us to both interpret the reanalysis data and begin to separate different
factors affecting the paleoclimate. In the case of Tambora, we identified the possible short-term effects of a low-
pressure system over western Europe, which may well have exacerbated the volcanic cooling in 1816, while
leading to warming in other areas. We also showed how ClimeApp's correlation and regression functions can be
used to combine ModE-RA with independent data, identifying statistical relationships between climatic and non-
climatic variables - in this case, volcanic forcing and Swiss bread prices.

ClimeApp, created and developed by historians, geographers and climate scientists, is as good an example as any
of the value of bridging science and humanities to advance interdisciplinary research. In this paper we have
focussed on the app's potential in our respective fields, but possible applications could be imagined in many of
the social and physical sciences. Our hope is that researchers from various disciplines will benefit from using
ClimeApp and ModE-RA, finding innovative and enlightening ways to integrate climate data into their own
research.

**Appendices**

**Appendix A: R libraries used by ClimeApp**

library(shiny)

library(ncdf4)

library(maps)

library(shinyWidgets)

library(RColorBrewer)

library(shinyjs)

library(bslib)

library(readxl)

library(xlsx)

library(DT)

library(zoo)

library(colourpicker)

library(tmaptools)

library(ggplot2)

library(sf)

library(shinylogs)

library(shinycssloaders)


ClimeApp v1.0 uses R-version 4.3.2.

**Appendix B: ClimeApp functions and data processing**

Behind the Shiny interface, the processing and analysis done by ClimeApp is relatively straightforward. ClimeApp utilises the set of R libraries in Appendix A to extract and process the raw ModE data into a format

selected by the user.

**Anomalies**

The anomaly map function shows the spatial distribution of climate anomalies averaged over a user-selected year range and month range. For example, June, July, August (JJA), 1501 to 1600 if your focus is boreal summer in the 16<sup>th</sup> century. The anomalies are created from 3 data products:

1. *Annual Means* – a timeseries of annual means for each point on the map, created by averaging absolute ModE values across the selected month range.

         2. *Reference Means* – a single reference mean for each point on the map, created by averaging *annual means* across a chosen reference year range.

         3. *Annual Anomalies* – a timeseries of annual anomalies for each point on the map, created by

subtracting the *reference means* from the *annual means*.

The final anomalies shown are the time-averaged annual anomalies. These are plotted using the base R plotting functions along with the coastlines and borders from the *maps* package. The anomaly timeseries is generated by averaging the annual anomalies for each year.

For reference, the calculations behind each data product are as follows:

The *annual mean* for a single year and single point on the map is given by the equation

$$AnnualMean = \overline{AbsoluteValues(M)}$$

where *M* is the selected month range.


The *reference mean* for a single year and point is given by

$$ReferenceMean = \overline{AnnualMeans(Y_{ref})}$$

where $Y_{ref}$ is the selected reference year range.

The *annual anomaly* for a single year and point is given by:

$$AnnualAnomaly = AnnualMean - ReferenceMean$$

Note that in the case of ModE-RAclim, the base data is already in anomaly format, so anomalies are merely calculated by subtracting time-averaged anomalies from each other.


The anomalies presented on the anomaly map and in the anomaly map data are given by

$$Anomaly(map) = \overline{AnnualMeans(Y)} - ReferenceMean = \overline{AnnualAnomalies(Y)}$$

where *Y* is the selected year range.

Anomalies presented on the timeseries map and timeseries data are given by

$$Anomaly(timeseries) = \overline{AnnualMeans(Lon, Lat)} - \overline{ReferenceMeans(Lon, Lat)}$$
$$= \overline{AnnualAnomalies(Lon, Lat)}$$

where *Lon* and *Lat* are the selected longitude and latitude range.

**Composites**

ClimeApp's composite maps show the time-averaged anomalies for a set of non-consecutive years, which can be entered or uploaded by the user. The anomaly reference period can be a fixed set of consecutive years, a custom set of non-consecutive years or an individual reference period generated for each year based on the *X* (a number of years chosen by the user) years prior. Calculations and plotting are performed in the same way as for anomalies, except for anomalies compared to *X* years prior

(XYP):
1.  *XYP Reference Means* – a set of reference means for each point on the map, one for each user-selected year. Calculated by averaging the *X* preceding *annual means*.
2.  *XYP Annual Anomalies* – a set of annual anomalies for each point on the map. Created by subtracting the corresponding *reference mean* from each *annual mean*.

To give an indication of the consistency of anomalies over the set of years in the composite, ClimeApp contains a '% sign match' statistical tool. This marks regions where the *annual anomalies* that form the composite agree in their sign more often than a user-defined threshold, given in percent. For example,

for a composite of five years, with anomalies of -1°C, -5°C, 1°C, 15°C and -3°C, the displayed mean would be a positive 1.4°C, but only 40% of the years would match this, since 3 are in fact negative.

**Correlation**

The correlation function allows users to generate a map of correlation coefficients, comparing either ModE variables or user-uploaded timeseries. Using the *cor()* function from the *stats* R package (R Core Team, 2022), it can employ either the Pearson or Spearman's Ranks correlation method. If both variables are in 'field' format, i.e. gridded map data, it performs a timeseries correlation of the *annual means* for each point on the map with the corresponding *annual means* for the second variable. If one variable is a timeseries however, it correlates each set of *annual means* with the same timeseries. In addition to the map, ClimeApp also produces a correlation timeseries, showing an annual timeseries of both variables (spatially averaged in the case of ModE variables) and a single correlation coefficient and p-value, calculated from those timeseries. The p-value shows the probability that the correlation was produced by random chance rather than an actual relationship between the variables. $p < 0.05$ is generally recommended for drawing legitimate conclusions.

**Regression**

In ClimeApp, regression operates in a similar way to correlation, performing a multiple linear regression analysis on a set of *annual means*. Using *lm()* from the *stats* R package, one or more independent variable timeseries are fitted to the dependent variable timeseries for each point on the map according to the model

$$V_{Dependent} = \beta_1 V_{Independent1} + \beta_2 V_{Independent2} + \cdots + \alpha + Residual$$

where $\beta$ is the coefficient and $\alpha$ is the intercept. ClimeApp then plots the spatial average of the dependent variable, trend ($\beta_1 V_{Independent\ 1} + \beta_2 V_{Independent\ 2} + \ldots$) and residual as a timeseries. Provided the dependent variable is a field, maps of the coefficients for each independent variable can be produced, as can maps of the p-values and residuals for each year.

**Annual Cycles**

This function shows the spatially averaged monthly ModE values over a given year or set of years. In the case of a set of years, these can be presented individually or as an average.

**Source Analysis and Further Statistical Functions**

The accuracy of ModE-RA is dependent on the availability and reliability of observations to constrain the model ensemble of ModE-Sim. To capture this, ClimeApp includes tools for visualizing the sources used to create ModE-RA and ModE-RAclim and the standard deviation (SD) ratio of the ModE-RA and ModE-Sim ensembles. The ModE-RA sources are presented as a semi-annual map showing the data points assimilated for each half-year, grouped by type and variable (see Figure 6). This allows the user to see where proxy, documentary or instrumental observations were integrated into the reconstruction and any gaps in the data. The SD ratio meanwhile, is the standard deviation of the ModE-Sim ensemble divided by the standard deviation of ModE-RA after the assimilation of observations:

$$SDratio = \frac{\sigma_{ModE-RAEnsemble}}{\sigma_{ModE-Ensemble}}$$

This gives a value between 0 and 1 for each month and grid point, with 1 showing no constraint (i.e. the ModE-RA output is the same as that of ModE-Sim and entirely generated from the models) and lower values showing increasing constraint by observations, meaning there are either more observations or that they are more 'trusted' by the reconstruction. The temporal mean of the SD ratio can be presented in ClimeApp as a contour map or grid-point overlay on the anomaly maps.

On timeseries plots, users have the option to add percentiles and moving averages. The moving averages are calculated using a rolling mean of timeseries values over a number of years selected by the user (default 11). To generate the percentiles, a Shapiro-Wilk test (Shapiro and Wilk, 1965) is first conducted on the timeseries data. If the data is normally distributed, which is rare for ModE timeseries,

then percentiles are calculated from the mean and standard deviation of the timeseries using the *qnorm()* function from the *stats* package. If the distribution is non-normal, ClimeApp instead finds the value corresponding to the quantile matching the users selection (i.e. for the 0.95 percentile, it returns values that 5% of all values are above/below), using the *quantile()* function from the *stats* package.

 **Appendix C: 1816 sea level pressure and 500hPa geopotential height anomalies**

Sea level pressure and 500hPa geopotential height anomalies for boreal summer (June to August, JJA) 1816, as compared to reference period 1799 to 1821. Showing results from ModE-RA - (1) & (3), and ModE-Sim - (2) & (4).

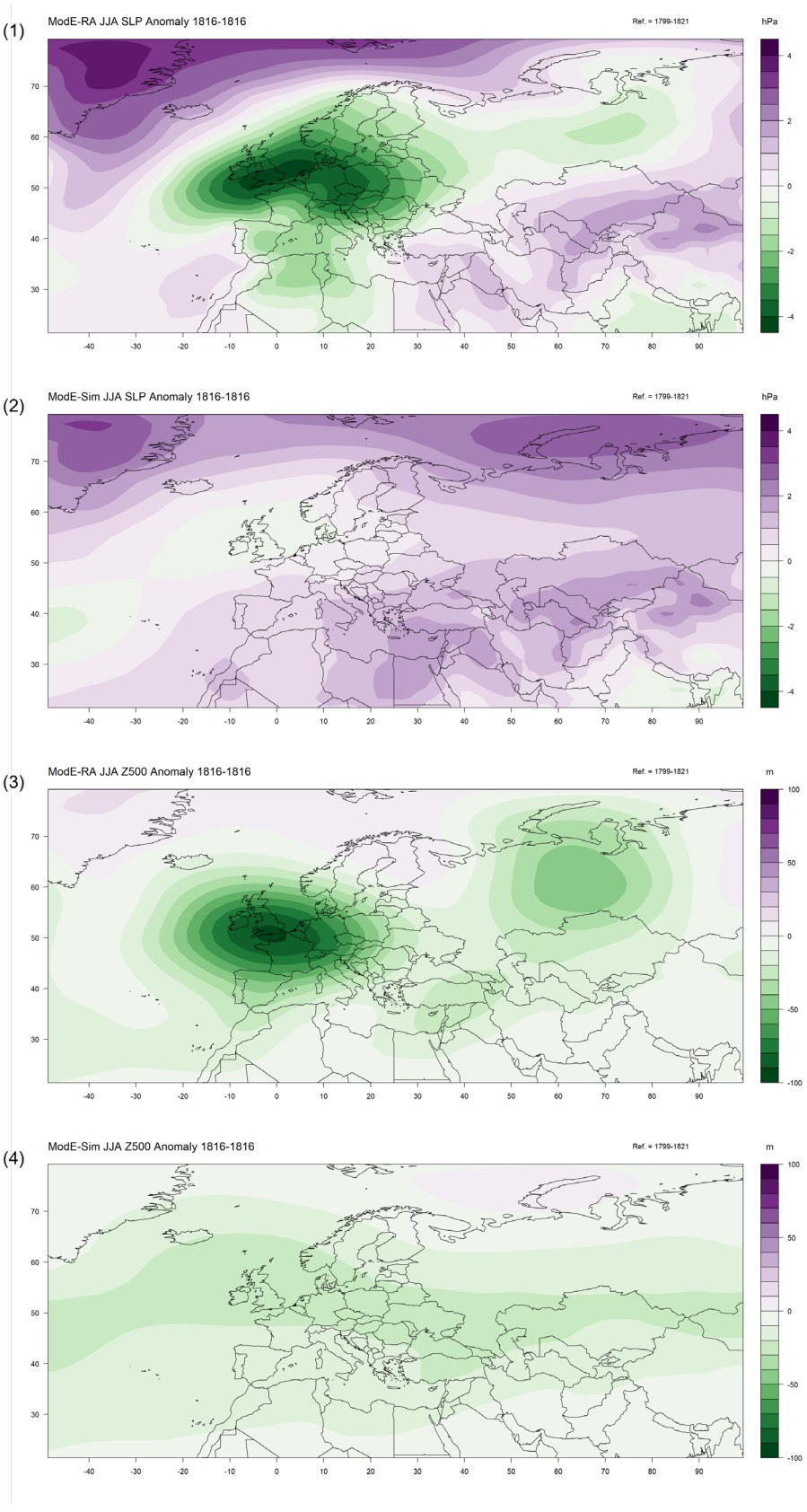

**Appendix D: Current limitations and future development**

The data available in ClimeApp is currently limited to ModE-RA, ModE-Sim and ModE-RAclim, and only four variables within those datasets. There are limited options for users to upload and process other data, but the format of this data is currently restricted to annual time series. The web app's ability to host a large number of users at the same time is also constrained by current processing power. To address some of these issues and to add further functionality to ClimeApp, a four-stage development plan has been devised:

| | |
|---|---|
| 1. Facilitation | • Interactive map to look at ModE-RA sources as an access point for the ModE-RA feedback archive, a database detailing each used source<br>• Several video tutorials to facilitate the use of ClimeApp<br>• Obtain funding and increase available processing power to reduce loading times and facilitate multiple simultaneous users |
| 2. Optimization | • Optimized and stream-lined plotting (e.g. various projections for global maps, pacific centred plotting)<br>• More options for customization (e.g. for regression and correlation)<br>• Ability to export georeferenced raster and vector files for usage in GIS |
| 3. Implementation | • Access to individual ModE-Sim ensemble members for more detailed study<br>• Access to more variables from ModE-RA, such as wind speed and direction<br>• Possibility for users to upload their own georeferenced data for purposes of plotting, averaging and correlation/regression against the ModE-RA data |
| 4. Cooperation | • API for other web-based research environments such as nodegoat (van Bree and Kessels, 2013) |

**Appendix E: Experience documentation, outreach and feedback**

ClimeApp underwent extensive internal and external testing before being deployed for general use. The beta version was introduced during a summer school in summer 2023 (Huhtamaa and Hibberts, 2024) and later trialled at several workshops and conferences, with both historians and climatologists. The initial user interactions and resulting valuable feedback included (but was not limited to): The addition of help texts to explain the different functions and options; the implementation of an interactive map of all ModE-RA sources to allow detailed source evaluation; faster data processing; more options for composite analysis; the addition of reference maps to see the reference and absolute values for anomalies; customization options for plots; and facility for multiple simultaneous users. This feedback was documented and either implemented or road mapped for future development (see Appendix D).

The current version of ClimeApp has already been included in the curriculum of two courses at the University of Bern: Brönnimann, 2023, Climatology III (Climate variability and change); and Huhtamaa, 2023, Climate and Society in History. It will also be included in further courses at the University of Bern from 2024 onwards including a four-session workshop introducing ClimeApp and its applications.

The full application was launched in early 2024 and presented at the EGU conference in Vienna and the Climate
of the Past and Societal Responses to Environmental Changes conference in Bern.

ClimeApp has its own feedback and suggestion email address (climeapp.hist@unibe.ch), presented on the *Welcome* page, where users can report any issues and suggest improvements. These are then considered by the developers before being added to the application's Trello page (https://trello.com/b/3hKu3RlL/climeapp-development), where users to track their suggestions and see what we are currently working on. These
550 contributions are vital for making ClimeApp as useful and user-friendly as possible.

## Code availability

The essential code of ClimeApp – *app.R* and *helpers.R* – is available on the projects GitHub page. https://github.com/ClimeApp/ClimeApp_development

## Data availability

The Mode-RA database can be downloaded at: https://www.palaeo-ra.unibe.ch/data_access/

## Author contributions

NB and RW conceptualized, created and developed ClimeApp as part of their respective PhD projects. They conducted research and analysed data for the case study, produced all graphs and figures and contributed equally to the manuscript. NW co-developed ClimeApp, is responsible for the mail-feedback and manages the Trello
page. JF and RH wrote the SD ratio function used in ClimeApp, set up the university server and gave technical advice on the ModE data and contributed to the writing of the manuscript. HH and SB encouraged and promoted the creation of the application and helped with the scientific outreach for the project. All authors discussed the project and its results, helped with the quality control of the final product and provided comments on the manuscript.

**Competing Interests**

The authors declare that they have no conflict of interest.

## Disclaimer

## Acknowledgements

The ECHAM6 simulations (ModE-Sim) were performed at the Swiss Supercomputer Centre (CSCS) and the
570 University of Bern Linux Cluster (UBeLix).

Luc Billaud: Graphics designer and creator of the ClimeApp logo.

**Financial Support**

NB, RW, NW and HH were supported by the Swiss State Secretariat for Education, Research and Innovation (contract number MB22.00030) and Swiss National Science Foundation (grant number PZ00P1_201953). JF, RH and SB received funding from the European Union's Horizon 2020 (grant number 787574).

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
