# Peer review of "ClimeApp: Data processing tool for monthly, global climate data from the ModE-RA paleo-reanalysis, 1422 to 2008 CE"

_EGUsphere, 2024_

## Author Response (AR1)

**Paper Title:**

**ClimeApp: Opening Doors to the Past Global Climate. New Data Processing Tool for the ModE-RA Climate Reanalysis**

**Action taken in response to reviewer comment RC1:**

(*Comment* / Action)

*Summary: The manuscript presents software to access data sets that may be interesting for researchers of the paleoclimate of the past centuries. This data set comprises three 'reconstructions' of seasonal climate fields based on climate simulations and an off-line assimilation scheme that merged the output of these simulations with natural proxies and long instrumental records.*

*Recommendation: I have some recommendations to improve the clarity of the text. In some instances, the text is indeed rather unclear, for instance, in the description of the differences between the three different data sets. To really understand these differences, I needed to look up the Valler et al. (2024) paper, and I think that this manuscript should provide enough clear information for the interested reader without the need to look up the original publications. Other than these recommendations, I think that the manuscript and the software are a relevant contribution to palaeoclimate research, and it will facilitate the use of these data sets by other groups.*

*1) Title: I found the title too 'literary'. This title would be fine for an internet site or a press release, but not really for a research paper. In its present form, it is not informative, and it should include specifications of the time scale, type of tool, and spatial extent of the data. I suggest including ModE-RA Climate Reanalysis, webtool, global, past centuries, and seasonal in the title and keeping the title technical.*

> Addressed. See lines 1 & 2 in the tracked change document.

*2) 'ClimeApp allows access to the ModE-Sim climate simulation, which is the basis of ModE-RA before assimilating early instrumental, documentary and proxy data.'*

*Actually, ClimApp allows access to all three data sets. This sentence may confuse the reader.*

> Addressed. See Abstract, lines 18 – 21 in the tracked change document.

*3) 'ModE-Sim is a climate model experiment'*

*ModE-Sim is not really a climate model experiment, and this terminology can be confusing for tan average paleoclimate reader - Please, keep the expected reader in mind (!). ModE-Sim is an ensemble of global climate simulations driven by external forcings.*

> Addressed. See Section 2.1, line 70 in the tracked change document.

*4) Originally designed to form the physical basis for ModE-RA,*

*This sentence may be unclear to the average reader. Observations also form 'a physical basis', so it can be argued that MedE_Sim and the observations both are the physical basis for ModE-RA*

 Addressed. See Section 2.1, line 70 in the tracked change document.

*5) 'The ModE-Sim ensemble mean used in ClimeApp represents the average over a set of climate states (the "ensemble members") that the model assumes to be realistic given the external forcings and boundary conditions.'*

*Consider a clearer version of this sentence, for instance: Each member in the ModE-Sim ensemble represents a possible climate state that is compatible (from the model's perspective) with the external forcing). The ensemble mean is the average over all ensemble members.*

 Addressed. See Section 2.1, lines 75-77 in the tracked change document.

*6) 'Averaging reduces temporal variability in the ensemble mean, compared with observations, but retains and highlights signals caused by variations in the forcings and boundary conditions, e.g. the climate's reaction to a volcanic eruption. '*

*Averaging over the ensemble members also reduces the spatial variability, not only the temporal variability.  The original sentence is, in my opinion, correct, but it may mislead the reader. Also, consider replacing boundary conditions by specifying SSt and sea-ice. This will help the average paleoclimatologist.*

 Addressed. See Section 2.1, lines 79-81 in the tracked change document.

*7) ' ModE-RA it can also help climatologists identify how observations affect the final reanalysis.'*

*This sentence, and actually the description of ModE-Clim is rather cryptic.*

 Addressed. See Section 2.1, line 92 and Section 2.3, lines 118-124 in the tracked change document.

*8) 'with observations increasing exponentially through time. Starting from a few thousand natural proxies and historical documents in the 15th century, by the late 19th century approximately 100000 mostly instrumental measurements are assimilated each year.'*

*Exponentially ? I do not tink this is the case. Probably, the authors mean increasing very rapidly - until they reach saturation.*

 Addressed. See Section 2.2, lines 100-102 in the tracked change document.

*9) ' this allows accurate reconstruction of the autumn, winter and spring seasons, in addition to the widespread tree-ring based summer reconstructions.'*

*In principle, the setup allows for a seasonal reconstruction. Whether or not the reconstruction is accurate is another matter.*

 Addressed. See Section 2.2, lines 105-108 in the tracked change document.

10) ' The current resolution for ModE-RA is 1.875° (longitude) by 1.865° (latitude)'

spatial horizontal resolution

  Addressed. See Section 2.2, lines 113-114 in the tracked change document.

11) ' This means that in ModE-RAclim, the externally forced signal in the model simulations is removed from the ensemble and only added back if it appears in the observations. '

As noted before, I found the description of Mod-E-RAclim rather confusing, and I needed to go back to the original paper by Valler et al. to really understand the difference. If I am not mistaken, the difference between ModE-RA and ModE-RAClim is the construction of the prior. For ModE-RA, the prior is constructed from the time-aligned ensemble members of Mod-Sim, i.g. the prior for the year 1800 is constructed from all simulated states for that particular year. For ModE-RAClim the prior is constructed from temporally non-aligned simulated states, e.g. the prior for 1800 includes all years of the ensemble ModE-Sim, regardless of the simulated year. Is my interpretation correct? If so, please spare a few lines to describe it more clearly. If not, please consider describing the ModE-RAClim in a much more detailed manner.

In my interpretation, the model error-covariance (spread) for ModE-RAClim is generally larger than for Mod-RA. For this reason, the impact of assimilating observations in ModE-RAClim is stronger. Please confirm if this is correct.

  Addressed. See Section 2.3, lines 118-124 in the tracked change document.

**Action taken in response to reviewer comment RC2:**

(*Comment* / Action)

*Dear authors of the manuscript titled "ClimeApp: Opening Doors to the Past Global Climate New Data Processing Tool for the ModE-RA Climate Reanalysis", the development of ClimeApp is a significant advancement in making paleoclimate reanalysis data more accessible. The application adeptly bridges climatology with humanities and other non-climatological disciplines, fulfilling a crucial interdisciplinary need. The manuscript commendably elucidates the technical underpinnings and functionalities of ClimeApp, offering a detailed exposition of features such as anomaly mapping, compositing, and statistical analyses. Additionally, the use of the Tambora eruption case study effectively showcases ClimeApp's utility in deriving new insights from historical climate events, thereby demonstrating its practical application and value.*

*However, the manuscript could be enhanced in the following ways:*

*Comparative Analysis: A comparative study between ClimeApp and other existing tools in paleoclimate research would enrich the manuscript. Such an analysis should highlight ClimeApp's unique features and advancements, further substantiating its contribution to the field.*

  Addressed. See section 5.1, lines 369-391 in the tracked change document.

*Methodological Detailing: The manuscript could benefit from more detailed explanations in certain sections to enhance reader understanding. Specifically, a more comprehensive description of how ClimeApp differentiates between external forcing and internal variability through its statistical or computational methods would be beneficial.*

Addressed. See section 2.1, lines 77-81 and section 2.2, line 109-111 in the tracked change document.

*User Experience Documentation: While the paper provides an in-depth description of ClimeApp's interface functionalities, it lacks empirical data from user feedback or usability studies. Including findings from beta testing or initial user interactions would lend credence to the claims regarding the app's user-friendliness and effectiveness.*

Partially addressed. See Appendix 4, lines 491-499 in the tracked change document.

*Future Development Roadmap: The discussion regarding future enhancements and the expansion potential of ClimeApp is intriguing yet lacks specificity. Detailing forthcoming features, enhancements, and a clear development roadmap, particularly concerning scalability issues like handling larger datasets or increased user traffic, would provide a clearer picture of the app's growth prospects.*

Addressed. See Appendix 3, lines 487-489 in the tracked change document.

---

## Author Response (AR2)

**Paper Title:**

**ClimeApp: Data processing tool for monthly, global climate data from the ModE-RA paleo-reanalysis, 1422 to 2008 CE**

**Response to Report #1 and action taken:**

(*Comment* / Response and action taken)

*Dear authors of the manuscript "ClimeApp: Data processing tool for monthly, global climate data from the ModE-RA paleo-reanalysis, 1422 to 2008 CE", I appreciate the substantial effort put into revising this manuscript. The current version is clearer and emphasizes the influence of the Tambora volcanic eruption effectively. However, I still have concerns about the scientific significance. Therefore, I suggest that the manuscript be accepted for publication after minor revisions.*

> Thankyou very much for your comments and suggestions. We hope we have addressed each point, as shown below:

*Main Comments:*
*1. I recommend including at least one new finding or result regarding the Tambora volcanic eruption to highlight its scientific significance. For example, explain how to understand the warming after the Tambora volcanic eruption, which is not consistent with the simulated results.*

> Accepted. We have added in a more detailed analysis of the possible non-volcanic temperature changes in 1816, supported by additional sea level pressure and 500 hPa geopotential height plots in the supplementary material.
>
> See Section 4.1., lines 212-216, Section 4.3, lines 291-293, and Section 6, lines 383-385 in the revised track-changes document and Appendix C in the revised supplementary material.

*2. The ModE-Sim simulation does not consider the atmosphere-ocean coupling, which should be mentioned, since this may have a systematic error.*

> Accepted. See Section 2.1, lines 70-73 in the revised track-changes document.
>
> You are correct that there is no two-way coupling in the model, but volcanic signals may be implicitly (partly) contained in the SST forcing.

*3. Add a section in the discussion to explain the Tambora eruption for interdisciplinary research.*

> Accepted. See Section 4, lines 186-196 in the revised track-changes document.
>
> As mentioned in the introduction, the Tambora eruption is indeed a prime example for interdisciplinary research. This has been now better highlighted also in the main body

as scientists from all disciplines (such as history, climatology, aerosol science, geology, epidemiology collaborated to fully comprehend its mechanisms and impacts. This development has its roots already in the 19th century.

*Specific Comments:*
*1. Regarding the difference between ModE-RA and ModE-RAclim, can it be considered online assimilation or offline assimilation?*

Both can be considered offline, i.e. the simulations are run completely separated from the assimilation. The difference is prior/model ensemble before the assimilation is conducted. In ModE-RA this prior are transient model simulations, which are in agreement with external forcings. In case of ModE-RAclim the prior are random years and ensemble members, i.e. just physically plausible climate patterns but not necessarily in agreement with external forcings.

We have updated Section 2.3, line 97 in the revised track-changes document to hopefully make this clear.

*2. In the introduction, could you briefly outline the current issues faced in the study of Tambora volcanic activity, such as the lack of comprehensive interdisciplinary research?*

Accepted. See Section 1, lines 31-39 in the revised track-changes document.

There is an emphasis in the recent research for the necessity of interdisciplinary approaches looking at the impacts of the Tambora eruption. This has been highlighted in the introduction.

*3. Line 57: What is the specific meaning of "sea ice acting"?*

This refers to the fact that sea ice (and the sea surface temperatures) are used as boundary conditions for the simulations. We have edited this slightly to make this clearer – see Section 2.1, line 65 in the revised track-changes document.

*4. How do you distinguish the seasonal features using most of the proxy records with annual resolution?*

In the earlier period, where annual proxies dominate the input data source, the model fields are only updated/corrected in the site specific season, which each proxy represents (e.g. Alpine tree-ring width could best represent JJAS temperature). Details are described in Valler et al. 2024. Fortunately, at the time of the Tambora eruption there are many monthly resolved observations in Europe, so the reanalysis does not rely solely on these proxies

*5. Line 183: Specify that it is for summer (June to August, JJA).*

Accepted. See Section 4.1, line 204 in the revised track-changes document.

*6. Lines 219-220: A figure of direct comparison is a good idea to show the consistency between the results and previous reconstructions.*

Unfortunately, we were not able to include the original figures from Wegmann et al., 2014 or Brönnimann, 2015. However, we have now directly referenced the original figures to allow for easier comparison.

See Section 4.1, lines 249-252 in the revised track-changes document.

*7. Figure 8c: I am curious about what leads to the positive pattern.*

We believe this may be due to additional internal variation, not related to the volcanic eruption.

See Section 4.1, lines 216-222 and Section 4.3, lines 291-293 in the revised track-changes document for an updated explanation.

*8. Line 298: "though however" is not appropriate.*

Accepted. This now reads "however" - see Section 5.1, line 332 in the revised track-changes document.

*9. Line 309: "very latest reanalysis data in Climate Explorer" is not true.*

Accepted. We have edited this to only the ModE-RA data - see Section 5.1, line 343 in the revised track-changes document.

*10. In the conclusion, explain the new information about the Tambora volcanic eruption and highlight the significance of this APP for scientific research.*

Accepted. We have updated the conclusion to highlight the significance of the app and the case study, see Section 6, lines 380-388 in the revised track-changes document.